

# Development of vocal emotion recognition in school-age children: The EmoHI test for hearing-impaired populations

Leanne Nagels[1,2], Etienne Gaudrain[2,3], Deborah Vickers[4], Marta Matos Lopes[5,6], Petra Hendriks[1] and Deniz Başkent[2]

[1] Center for Language and Cognition Groningen (CLCG), University of Groningen, Groningen, The Netherlands
[2] Department of Otorhinolaryngology/Head and Neck Surgery, University Medical Center Groningen, Groningen, The Netherlands
[3] CNRS, Lyon Neuroscience Research Center, Université de Lyon, Lyon, France
[4] Cambridge Hearing Group, Clinical Neurosciences Department, University of Cambridge, Cambridge, United Kingdom
[5] Hearbase Ltd, The Hearing Specialists, Kent, United Kingdom
[6] The Ear Institute, University College London, London, United Kingdom

Corresponding author
Leanne Nagels, leanne.nagels@rug.nl

## ABSTRACT

Traditionally, emotion recognition research has primarily used pictures and videos, while audio test materials are not always readily available or are not of good quality, which may be particularly important for studies with hearing-impaired listeners. Here we present a vocal emotion recognition test with pseudospeech productions from multiple speakers expressing three core emotions (happy, angry, and sad): the EmoHI test. The high sound quality recordings make the test suitable for use with populations of children and adults with normal or impaired hearing. Here we present normative data for vocal emotion recognition development in normal-hearing (NH) school-age children using the EmoHI test. Furthermore, we investigated cross-language effects by testing NH Dutch and English children, and the suitability of the EmoHI test for hearing-impaired populations, specifically for prelingually deaf Dutch children with cochlear implants (CIs). Our results show that NH children's performance improved significantly with age from the youngest age group onwards (4–6 years: 48.9%, on average). However, NH children's performance did not reach adult-like values (adults: 94.1%) even for the oldest age group tested (10–12 years: 81.1%). Additionally, the effect of age on NH children's development did not differ across languages. All except one CI child performed at or above chance-level showing the suitability of the EmoHI test. In addition, seven out of 14 CI children performed within the NH age-appropriate range, and nine out of 14 CI children did so when performance was adjusted for hearing age, measured from their age at CI implantation. However, CI children showed great variability in their performance, ranging from ceiling (97.2%) to below chance-level performance (27.8%), which could not be explained by chronological age alone. The strong and consistent development in performance with age, the lack of significant differences across the tested languages for NH children, and the above-chance performance of most CI children affirm the usability and versatility of the EmoHI test.

## INTRODUCTION

Development of emotion recognition in children has been studied extensively using visual stimuli, such as pictures or sketches of facial expressions (e.g., *Rodger et al., 2015*), or audiovisual materials (e.g., *Nelson & Russell, 2011*), and particularly in some clinical groups, such as autistic children (e.g., *Harms, Martin & Wallace, 2010*). However, not much is known about the development of vocal emotion recognition, even in typically developing children (*Scherer, 1986*; *Sauter, Panattoni & Happé, 2013*). While children can recognize facial and vocal emotions reliably and associate them with external causes already from the age of 5 years on (*Pons, Harris & De Rosnay, 2004*), it seems to take until late childhood for this ability to develop to adult-like levels (*Tonks et al., 2007*; *Sauter, Panattoni & Happé, 2013*). The recognition of vocal emotions relies heavily on the perception of related vocal acoustic cues, such as mean fundamental frequency (F0) and intensity, as well as fluctuations in these cues, and speaking rate (*Scherer, 1986*). Based on earlier research on the development of voice cue perception (*Mann, Diamond & Carey, 1979*; *Nittrouer & Miller, 1997*), children's performance may be lower compared to adults due to differences in their weighting of acoustic cues and a lack of robust representations of auditory categories. For instance, *Morton & Trehub (2001)* showed that when acoustic cues and linguistic content contradict the emotion they convey, children mostly rely on linguistic content to judge emotions, whereas adults mostly rely on affective prosody. In addition, children and adults both are better at facial emotion recognition than vocal emotion recognition (*Nelson & Russell, 2011*; *Chronaki et al., 2015*). All of these observations combined indicate that the formation of robust representations for vocal emotions is highly complex and possibly a long-lasting process even in typically developing children.

Research with hearing-impaired children has shown that they do not perform as well on vocal emotion recognition compared to their normal-hearing (NH) peers (*Dyck et al., 2004*; *Hopyan-Misakyan et al., 2009*; *Nakata, Trehub & Kanda, 2012*; *Chatterjee et al., 2015*). *Hopyan-Misakyan et al. (2009)* showed that 7-year-old children with cochlear implants (CIs) performed as well as their NH peers on visual emotion recognition but scored significantly lower on vocal emotion recognition. Visual emotion recognition generally seems to develop faster than vocal emotion recognition (*Nowicki & Duke, 1994*; *Nelson & Russell, 2011*), particularly in hearing-impaired children (*Hopyan-Misakyan et al., 2009*), which may indicate that visual emotion cues are perceptually more prominent or easier to categorize than vocal emotion cues. For hearing-impaired children, a higher reliance on visual emotion cues as compensation for spectro-temporally degraded auditory input may be an effective strategy, as emotion recognition in daily life is usually multimodal. However, it may lead to less robust auditory representations of vocal emotions and knowledge about their acoustic properties. *Luo, Kern & Pulling (2018)* also showed that CI users' ability to recognize vocal emotions was related to their self-reported quality of

life, which emphasizes the importance of recognizing vocal emotion cues in addition to visual emotion cues. *Wiefferink et al. (2013)* suggested that reduced auditory exposure and language delays may also lead to delayed social-emotional development and reduced conceptual knowledge about emotions, which in turn result in a negative impact on emotion recognition. This is also evidenced by CI children's reduced differences in mean F0 cues and F0 variations in emotion production compared to their NH peers (*Chatterjee et al., 2019*). The effects of conceptual knowledge on children's discrimination abilities have also been shown earlier, for instance, in research on pitch discrimination (*Costa-Giomi & Descombes, 1996*). *Costa-Giomi & Descombes (1996)* showed that French children showed better pitch discrimination when they had to use the single meaning terms 'aigu' and 'grave' to denote pitch than the multiple meaning words 'haut' [high] and 'bas' [low], which besides pitch also can be used to denote differences in space and loudness. This finding demonstrates that the label to refer to a concept may affect the labeling process itself. Thus, if CI children do not have clear conceptual knowledge about emotions, this will also similarly affect their ability to label them correctly. Finally, perceptual limitations, such as increased F0 discrimination thresholds (*Deroche et al., 2014*), may also play a role in CI children's abilities to recognize vocal emotions. *Nakata, Trehub & Kanda (2012)* found that children with CIs especially had difficulties with differentiating happy from angry vocal emotions. This finding suggests that CI children primarily use speaking rate to categorize vocal emotions, as this cue differentiates sad from happy and angry vocal emotions but is similar for the latter two emotions. Therefore, hearing loss also seems to influence the weighting of different acoustic cues, and hence likely also affects the formation of representations of vocal emotions.

Vocal emotion recognition also differs from visual emotion recognition due to the potential influence of linguistic factors. Research regarding cross-language effects on emotion recognition has also demonstrated the importance of auditory exposure for vocal emotion recognition. Most studies have demonstrated a so-called 'native language benefit' showing that listeners are better at recognizing vocal emotions produced by speakers from their own native language than from another language (*Van Bezooijen, Otto & Heenan, 1983*; *Scherer, Banse & Wallbott, 2001*; *Bryant & Barrett, 2008*). This effect has been mainly attributed to cultural differences (*Van Bezooijen, Otto & Heenan, 1983*), but also effects of language distance have been reported (*Scherer, Banse & Wallbott, 2001*), i.e., differences in performance were larger when the linguistic distance (the extent to which the features of two languages differ from each other) between the speakers' and listeners' native languages was larger. Interestingly, *Bryant & Barrett (2008)* did not find a native language benefit for low-pass filtered vocal emotion stimuli, which filtered out both the linguistic message and the language-specific phonological information. *Fleming et al. (2014)* also demonstrated a similar native-language benefit for voice recognition based on differences in phonological familiarity. For CI children, reduced auditory exposure may also lead to reduced phonological familiarity, and therefore also contribute to difficulties with the recognition of vocal emotions.

As most research on the development of emotion recognition has used visual or audiovisual materials such as pictures or videos, good-quality audio materials are scarce.

While the audio quality may only have a small effect on NH listeners' performance, it may be imperative for hearing-impaired listeners' vocal emotion recognition abilities. Hence, we recorded high sound quality vocal emotion recognition test stimuli produced by multiple speakers with three basic emotions (happy, angry, and sad) that are suitable to use with hearing-impaired children and adults: the EmoHI test. We aimed to investigate how NH school-age children's ability to recognize vocal emotions develops with age and to obtain normative data for the EmoHI test for future applications, for instance, with clinical populations. In addition, we tested children of two different native languages, namely Dutch and English, to investigate potential cross-language effects, and we collected preliminary data from Dutch prelingually deaf children with CIs, to investigate the applicability of the EmoHI test to hearing-impaired children.

## MATERIALS & METHODS

### Participants

We collected normative data from fifty-eight Dutch and twenty-five English children between 4 and 12 years of age, and fifteen Dutch and fifteen English adults between 20 and 30 years of age with normal hearing. All NH participants were monolingual speakers of Dutch or English and reported no hearing or language disorders. Normal hearing (hearing thresholds at 20 dB HL) was screened with pure-tone audiometry at octave-frequencies between 500 and 4,000 Hz. In addition, we collected preliminary data from fourteen prelingually deaf Dutch children with CIs between 4 and 16 years of age. The study was approved by the Medical Ethical Review Committee of the University Medical Center Groningen (METc 2016.689). A written informed consent form was signed by adult participants and the parents or legal guardians of children before data collection.

### Stimuli and apparatus

We made recordings of six native Dutch speakers producing two non-language specific pseudospeech sentences using three core emotions (happy, sad, and angry), and a neutral emotion (not used in the current study). All speakers were native monolingual speakers of Dutch without any discernible regional accent and did not have any speech, language, or hearing disorders. Speakers gave written informed consent for the distribution and sharing of the recorded materials. To keep our stimuli relevant to emotion perception literature and suitable for usage across different languages, the pseudospeech sentences that we used, *Koun se mina lod belam* [kʌun sə mina: lɔd be:lam] and *Nekal ibam soud molen* [ne:kal ibam sʌut mo:lən], were based on the Geneva Multimodal Emotion Portrayal (GEMEP) Corpus materials by *Bänziger, Mortillaro & Scherer (2012)*. These pseudosentences are meaningful neither in Dutch nor in English, nor in any other Indo-European languages. Speakers were instructed to produce the sentences in a happy, sad, angry, or neutral manner using emotional scripts that were also used for the GEMEP corpus stimuli (*Scherer & Bänziger, 2010*). We chose these three core emotions as previous studies have reported that children first learn to identify happy, angry, and sad emotions, respectively, followed by fear, surprise, and disgust (*Widen & Russell, 2003*), and hence we could test children

**Table 1** Overview of the EmoHI test speakers' voice characteristics.

| Speaker | Age (years) | Gender | Height (m) | Mean F0 (Hz) | F0 range (Hz) |
|---------|-------------|--------|------------|--------------|----------------|
| T2 | 36 | F | 1.68 | 302.23 | 200.71–437.38 |
| T3 | 27 | M | 1.85 | 166.92 | 100.99–296.47 |
| T5 | 25 | F | 1.63 | 282.89 | 199.49–429.38 |
| T6 | 24 | M | 1.75 | 167.76 | 87.46–285.79 |

from very young ages. The stimuli were recorded in an anechoic room at a sampling rate of 44.1 kHz.

We pre-selected 96 productions, including neutral productions, (2 productions × 2 sentences × 4 emotions × 6 speakers) and performed a short online survey with Dutch and English adults to confirm that the stimuli were recognized reliably and to select the four speakers whose productions were recognized best. Table 1 shows an overview of these four selected speakers' demographic information and voice characteristics. The neutral productions and the productions of the other two speakers were part of the online survey, and are available with the stimulus set, but were not used in the current study to simplify the task for children. Our final set of stimuli consisted of 36 experimental stimuli with three productions (one sentence repeated once + the other sentence) per emotion and per speaker (3 productions × 3 emotions × 4 speakers) as well as 4 practice stimuli with one production per speaker that were used for the training session.

## Procedure

NH and CI children were tested in a quiet room at their home, and NH adults were tested in a quiet testing room at the two universities. Since the present experiment was part of a larger project on voice and speech perception (Perception of Indexical Cues in Kids and Adults (PICKA)), data were collected from the same population of children and adults in multiple experiments, see, for instance, *Nagels et al. (2020)*. The experiment started with a training session consisting of 4 practice stimuli and was followed by the test session consisting of 36 experimental stimuli. The total duration of the experiment was approximately 6 to 8 min. All stimuli were presented to participants in a randomized order.

The experiment was conducted on a laptop with a touchscreen using a child-friendly interface that was developed in Matlab (Fig. 1). The auditory stimuli were presented via Sennheiser HD 380 Pro headphones for NH children and adults, and via Logitech Z200 loudspeakers for CI children. The presentation level of the stimuli was calibrated to a sound level of 65 dBA. CI children were instructed to use the settings they most commonly use in daily life and to keep the settings consistent throughout the experiment. In each trial, participants heard a stimulus and then had to indicate which emotion was conveyed by clicking on one of three corresponding clowns on the screen. Visual feedback on the accuracy of responses was provided to motivate participants. Participants saw confetti falling down the screen after a correct response, and the parrot shaking its head after an incorrect response. After every two trials, one of the clowns in the back went one step up the ladder until the experiment was finished to keep children engaged and to give an indication of the progress of the experiment.

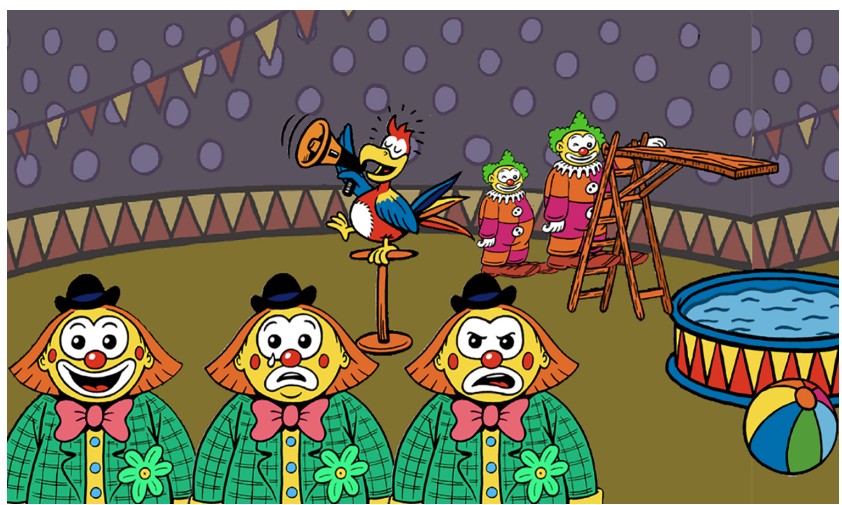

**Figure 1** **The experimental interface of the EmoHI test.** The illustrations were made by Jop Luberti. This image is published under the CC BY NC 4.0 license (https://creativecommons.org/licenses/by-nc/4.0/).

## Data analysis

NH children's accuracy scores were analyzed using the lme4 package (*Bates et al., 2014*) in R. A mixed-effects logistic regression model with a three-way interaction between *language* (Dutch and English), *emotion* (happy, angry, and sad), and *age* in decimal years, and random intercepts per participant and per stimulus was computed to determine the effects of language, emotion, and age on NH children's ability to recognize vocal emotions. We used backward stepwise selection with ANOVA Chi-Square tests to select the best fitting model, starting with the full factorial model, in lme4 syntax: `accuracy ~ language * emotion * age + (1 |participant) + (1 |stimulus)`, and deleting one fixed factor at a time based on its significance. In addition, we performed Dunnett's tests on the NH Dutch and English data with *accuracy* as an outcome variable and *age group* as a predictor variable using the DescTools package (*Signorell et al., 2018*) to investigate at what age NH Dutch and English children show adult-like performance. Finally, we examined our preliminary data of CI children to investigate if they could reliably perform the task.

## RESULTS

### NH Dutch and English data

Figure 2 shows the accuracy scores of NH Dutch and English participants as a function of their age (dots) and age group (boxplots). Model comparison showed that the full model with random intercepts per participant and per stimulus was significantly better than the full models with only random intercepts per participant [$\chi^2(1) = 393$, $p < 0.001$] or only random intercepts per stimulus [$\chi^2(1) = 51.9$, $p < 0.001$]. Backward stepwise selection showed that the best fitting and most parsimonious model was the model with only a fixed effect of *age*, in lme4 syntax: `accuracy ~ age + (1 |participant) + (1 |stimulus)`. This model did not significantly differ from the full model [$\chi^2(10) = 12.90$, $p = 0.23$] or any of
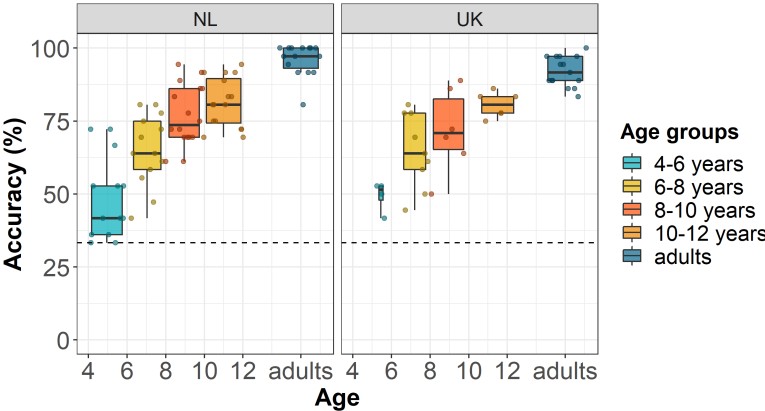

**Figure 2 Emotion recognition in NH children and adults.** Accuracy scores of NH Dutch and English children and adults for the EmoHI test per age group and per language. The dotted line shows the chance-level performance of 33.3% correct. The dots show individual data points at participants' age (Netherlands (NL): $N_{children} = 58$, $N_{adults} = 15$; United Kingdom (UK) : $N_{children} = 25$, $N_{adults} = 15$). The boxplots show the median accuracy scores per age group, and the lower and upper quartiles. The whiskers indicate the lowest and highest data points within plus or minus 1.5 times the interquartile range.

the other models while being the most parsimonious. Figure 2 shows the data of individual participants and the median accuracy scores per age group for the NH Dutch and English participants. NH children's ability to recognize vocal emotions correctly significantly increased as a function of age ($z$-value = 8.91, estimate = 0.30, SE = 0.034, $p < 0.001$). We did not find any significant effects of language or emotion on children's accuracy scores. Finally, the results of the Dunnett's tests showed that the accuracy scores of Dutch NH children of all tested age groups differed from Dutch NH adults (4–6 years difference = −0.47, $p < 0.001$; 6–8 years difference = −0.31, $p < 0.001$; 8–10 years difference = −0.19, $p < 0.001$; 10–12 years difference = −0.15, $p < 0.001$), and the accuracy scores of English NH children of all tested age groups differed from English NH adults (4–6 years difference = −0.43, $p < 0.001$; 6–8 years difference = −0.27, $p < 0.001$; 8–10 years difference = −0.20, $p < 0.001$; 10–12 years difference = −0.12, $p < 0.01$). The mean accuracy scores per age group and language are shown in Table 2.

## Preliminary data of CI children
Figure 3 shows the accuracy scores of Dutch CI children as a function of their chronological age (left panel) and hearing age (right panel), the latter based on the age at which they received the CI. The mean accuracy scores per age group are shown in Table 2. All except one CI child performed at or above chance-level. Based on Fig. 3, we can see that 7 out of 14 CI children (50%) performed within the NH age-appropriate range. Considering CI children's hearing age instead of their chronological age, 9 out of 14 CI children (64.3%) still show performance within the NH age-appropriate range. However, there is a large variability in CI children's performance, which varies from ceiling (97.2%) to below chance-level performance (27.8%). The development in CI children's performance with

**Table 2** Overview of the mean accuracy scores (%) per participant and age group.

| Age groups | Participant groups | | |
| --- | --- | --- | --- |
| | *Dutch NH* | *English NH* | *Dutch CI* |
| 4–6 years | 48.7% | 49.3% | 34.7% |
| 6–8 years | 65.2% | 64.8% | 48.6% |
| 8–10 years | 76.7% | 71.8% | 37.5% |
| 10–12 years | 81.2% | 80.6% | 57.6% |
| 12–14 years | – | – | 50.0% |
| 14–16 years | – | – | 76.9% |
| Adults | 96.1% | 92.0% | – |

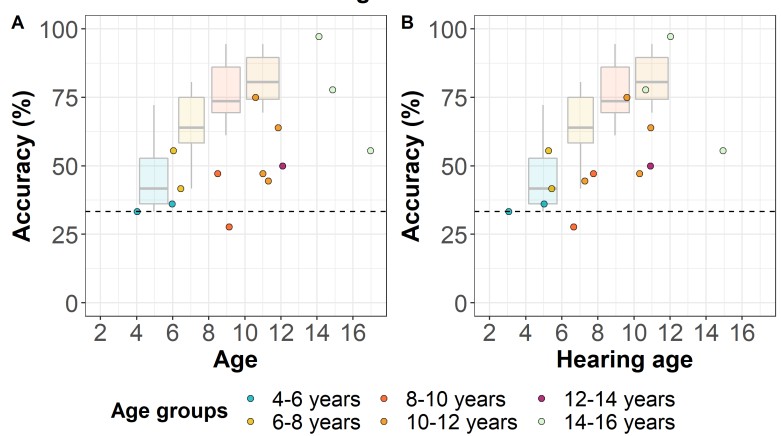

**Figure 3** **Emotion recognition in Dutch CI children.** Accuracy scores of Dutch CI children ($N = 14$) for the EmoHI test per age group. The dotted line shows the chance-level performance of 33.3% correct. The dots show individual data points at Dutch CI children's chronological age (A) and at their hearing age (B). The boxplots show NH Dutch children's median accuracy scores per age group, and the lower and upper quartiles, reproduced from Fig. 2. The whiskers indicate the lowest and highest data points of NH Dutch children within plus or minus 1.5 times the interquartile range.

age does not seem to be as consistent as we found for NH children, which suggests that their performance is not merely due to age-related development.

## DISCUSSION

### Age effect

As shown by our results and the data displayed in Fig. 2, NH children's ability to recognize vocal emotions improved gradually as a function of age. In addition, we found that, on average, even the oldest age group of 10- to 12-year-old Dutch and English children did not show adult-like performance yet. The 4-year-old NH children that were tested performed at or above chance level while adults generally showed near ceiling performance, indicating that our test covers a wide range of age-related performances. Our results are in line with previous findings that NH children's ability to recognize vocal emotions improves

gradually as a function of age (*Tonks et al., 2007*; *Sauter, Panattoni & Happé, 2013*). It may be that children require more auditory experience to form robust representations of vocal emotions or rely on different acoustic cues than adults, as was shown in research on the development of sensitivity to voice cues (*Mann, Diamond & Carey, 1979*; *Nittrouer & Miller, 1997*; *Nagels et al., 2020*). It is still unclear on which specific acoustic cues children are basing their decisions on and how this differs from adults. Future research using machine-learning approaches may be able to further explore such aspects. Finally, the visual feedback may have caused some learning effects, although the correct response was not shown after an error, and learning would pose relatively high demands on auditory working memory since there were only three productions per speaker and per emotion presented in a randomized order.

## Language effect

Comparing data from NH children from two different native languages, we did not find any cross-language effects between Dutch and English children's development of vocal emotion recognition, even though the materials were produced by Dutch native speakers. Earlier research has demonstrated that although adults are able to recognize vocal emotions across languages, there still seems to be a native language benefit (*Van Bezooijen, Otto & Heenan, 1983*; *Scherer, Banse & Wallbott, 2001*; *Bryant & Barrett, 2008*). Listeners were better at recognizing vocal emotions that were produced by speakers of their native language than another language. However, it should be noted that five (*Scherer, Banse & Wallbott, 2001*; *Bryant & Barrett, 2008*) and nine (*Van Bezooijen, Otto & Heenan, 1983*) different and more complex emotions were used in these studies which likely poses a considerably more difficult task than differentiating three basic emotions. In addition, the lack of a native language benefit in our results may also be due to the fact that Dutch and English are phonologically closely related languages. This idea is also in line with the language distance effect (*Scherer, Banse & Wallbott, 2001*) and phonological familiarity effects (*Bryant & Barrett, 2008*). We are currently collecting data from Turkish children and adults to investigate whether there are any detectable cross-language effects for typologically and phonologically more distinct languages.

## CI children

The preliminary data from the CI children show that only one CI child performed below chance-level, which shows that almost all CI children could reliably perform the task, and indicates that the task seems sufficiently easy to capture their vocal emotion recognition abilities. In addition, 7 out of 14 CI children performed within the NH age-appropriate range, and if we consider CI children's hearing age instead of their chronological age, 9 out of 14 CI children still fell within that range. Vocal emotion recognition performance was generally lower in CI children compared to NH children and did not seem to follow the same consistent improvement trajectory that we found for NH children. The general lower performance of CI children and the lack of a strong relation between CI children's performance and chronological or hearing age is in line with findings from previous studies (*Hopyan-Misakyan et al., 2009*; *Nakata, Trehub & Kanda, 2012*; *Chatterjee et al., 2015*). The

![PeerJ]

variability was large and covered the entire performance range, which also demonstrates that the EmoHI test can capture a wide range of performances. Besides age, CI children's performance seems to be heavily affected by differences in social-emotional development causing reduced conceptual knowledge on emotions and their properties (*Wiefferink et al., 2013*; *Chatterjee et al., 2019*), and differences in their hearing abilities causing perceptual limitations (*Nakata, Trehub & Kanda, 2012*). For instance, individual differences in CI children's vocal emotion recognition abilities may also rely on their F0 discrimination thresholds, which are generally higher and more variable in CI children compared to NH children (*Deroche et al., 2014*). We are currently working on an in-depth analysis of CI children's data, as their performance seems to also be largely related to their hearing abilities (*Nakata, Trehub & Kanda, 2012*), a perceptual effect, and social-emotional interaction and development (*Wiefferink et al., 2013*), a cognitive effect, in addition to age.

## CONCLUSIONS

The results of the current study provide baseline normative data for the development of vocal emotion recognition in typically developing, school-age children with normal hearing using the EmoHI test. Our results show that there is a large but relatively slow and consistent development in children's ability to recognize vocal emotions. Furthermore, the preliminary data from the CI children show that they seem to be able to carry out the EmoHI test reliably, but the improvement in their performance as a function of age was not as consistent as for NH children. The clear development observed in NH children's performance as a function of age and the generalizability of performance across the tested languages show the EmoHI test's suitability for different ages and potentially also across different languages. Additionally, the above-chance performance of most CI children and the high sound quality stimuli also demonstrate that the EmoHI test is suitable for testing hearing-impaired populations.

## ACKNOWLEDGEMENTS

We are grateful to all of the children, parents, and students that took part in the study, the speakers that were recorded for our stimuli, and Basisschool de Brink in Ottersum, Basisschool de Petteflet, and BSO Huis de B in Groningen for their help with recruiting NH child participants and the clinic of the Otorhinolaryngology Department of the University Medical Center Groningen, particularly ir. Bert Maat and Dr. Rolien Free, for their help with the recruitment of CI child participants. We would also like to thank Iris van Bommel, Evelien Birza, Paolo Toffanin, Jacqueline Libert, Jemima Phillpot, and Jop Luberti (illustrations) for their contribution to the development of the game interfaces, and Monita Chatterjee for her advice on recording the sound stimuli. Finally, we would like to thank the Vocal Interactivity in-and-between Humans, Animals, and Robots (VIHAR) workshop committee for awarding our proceedings paper with the PeerJ best contribution award which resulted in the current paper.

### Funding

This work was supported by the Center for Language Cognition Groningen (CLCG), a VICI Grant from the Netherlands Organization for Scientific Research (NWO) and the Netherlands Organization for Health Research and Development (ZonMw) (Grant No. 918-17-603), the Medical Research Council (Senior Fellowship Grant S002537/1), and by the framework of the LabEx CeLyA ("Centre Lyonnais d'Acoustique", ANR-10-LABX-0060/ANR-11-IDEX-0007), the French National Research Agency. The funders had no role in study design, data collection and analysis, decision to publish, or preparation of the manuscript.

### Grant Disclosures

The following grant information was disclosed by the authors:
Center for Language Cognition Groningen (CLCG).
VICI Grant from the Netherlands Organization for Scientific Research (NWO) and the Netherlands Organization for Health Research and Development (ZonMw): 918-17-603.
Medical Research Council: S002537/1.
LabEx CeLyA, French National Research Agency: ANR-10-LABX-0060/ANR-11-IDEX-0007.

### Competing Interests

Marta Matos Lopes is currently employed by Hearbase Ltd, The Hearing Specialists, Kent, United Kingdom.

### Author Contributions

- Leanne Nagels conceived and designed the experiments, performed the experiments, analyzed the data, prepared figures and/or tables, authored or reviewed drafts of the paper, and approved the final draft.
- Etienne Gaudrain conceived and designed the experiments, analyzed the data, authored or reviewed drafts of the paper, and approved the final draft.
- Deborah Vickers, Petra Hendriks and Deniz Başkent conceived and designed the experiments, authored or reviewed drafts of the paper, and approved the final draft.
- Marta Matos Lopes performed the experiments, authored or reviewed drafts of the paper, and approved the final draft.

### Human Ethics

The following information was supplied relating to ethical approvals (i.e., approving body and any reference numbers):

Ethical approval of the study was given by the Medical Ethical Review Committee of the University Medical Center Groningen (METc 2016.689).

### Data Availability

The raw data are available in the Supplemental Files. The NH children and adults data are available at: PICKA-NH: Vocal emotion recognition: Nagels, Leanne; Gaudrain, Etienne; Vickers, Deborah; Matos Lopes, Marta; Hendriks, Petra; Başkent, Deniz, 2020, "PICKA-NH: Vocal emotion recognition", https://hdl.handle.net/10411/WZVUGN, DataverseNL, V1.

The EmoHI test sound stimuli are on Zenodo: Nagels, Leanne, Hendriks, Petra, & Başkent, Deniz. (2020). The EmoHI Test stimuli: Measuring vocal emotion recognition in hearing-impaired populations. Zenodo. http://doi.org/10.5281/zenodo.3689710.

## Supplemental Information

Supplemental information for this article can be found online at http://dx.doi.org/10.7717/peerj.8773#supplemental-information.

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
