# Peer review of "Development of vocal emotion recognition in school-age children: The EmoHI test for hearing-impaired populations"

_PeerJ, doi:10.7717/peerj.8773_

## Round 0.1 · original submission · Minor Revisions

I have now received two reviews on your paper. As you will read, both reviewers find your paper interesting and appreciate your work. I share their opinion.

The reviews are very clear, so I invite you to carefully address the specific issues they raise. A few general comments that might be helpful for a successful revision:

First, I think you need to explain better why you decided to perform a cross-linguistic study, and to discuss your results further in the context of relevant literature (e.g. see for example cross-linguistic studies on pitch, e.g. Dolscheid et al. 2013).(see also comments of Reviewer 1)
Second, you should clarify the rationale underlying the choice of the emotions (see comments of Reviewer 2).

Finally, I think you need to better justify the sample size you choose, and why was the numerosity of the two groups so different.

·

Basic reporting

This study reports on normative data for vocal emotion recognition ability, between 4 and 12 years of age, as well as in adults. The emotions are restricted to 3 basic types: happy, sad, and angry, making it possible to perform even for kids as young as 4. Furthermore, participants were either Dutch or English speakers.

The results show no difference with emotion type, and no language differences, but an undeniable developmental trend (that did not interact with emotion or language). Surprisingly perhaps, the 12 y.o. children performed 15% more poorly than the adults, suggesting that development in emotional processing as a whole is far from complete in adolescence. Overall, this study is perfectly well designed, executed, and analyzed appropriately. It provides one more tool, the EmoHI test, to investigate emotional development across a wide range of ages and different populations of interest, which is much needed in the field. I have only a few minor comments to help improve the manuscript further.

Experimental design

The study was carefully designed and executed. There are only a few points of minor confusion.

Line 90-91: Can you clarify that these are non-intelligible sentences? But are they still sounding more like Dutch than English? Is that why you looked at two languages?

Line 100: I’m confused as to why there are three items. Each of the 4 speakers received the highest ratings online, on average, but in each speaker, you selected the highest-rated productions among several tokens of the same sentences? And this happened to be always 2 times one sentence and 1 time the other?

Line 119: seems like a cool game for kids, well done! Have you made this game available?

Validity of the findings

The findings make a lot of sense, and respond clearly to the goal of providing normative data in this task.

Line 152: “did not show adult-like performance yet”; indeed, and that’s quite impressive. Can you clarify the difference in percentage points here?

Line 166: Interesting. Could you say a few more lines on why this happens? Is it the type of F0 inflections that the native speakers use that sound very natural or more foreign?

Additional comments

Not much rationale is presented in the introduction as to why two different languages were examined. It may be useful to rearrange some elements from the discussion paragraph (162-173) back into the introduction.

A subtle point of discord: The authors state that CI children do not have any problems with identifying visual emotions (lines 51-53), implying that the deficits are exclusively driven by the auditory modality. This point is debatable as a function of age. In Hopyan-Misakyan’s study, the children were above seven years of age. At younger ages, this may not be quite true. For example, Wiefferink et al. (2012) looked at kids from 2.5 to 5, i.e. pre-school, and found more general impairment in emotion processing (in CI kids), including facial emotion identification. Fully understanding what an emotion entails, what it looks like from an outsider’s perspective, and what it feels like from an insider’s perspective, is a fairly complex thing to do. The more modalities are available, the easier it is to integrate the different aspects that characterize an emotion. For example, sadness is not just about eyes dropping and a lack of smile; it is equally important to appreciate that it generally sounds like a slow melodic pattern with fairly monotonous intonation and homogeneous intensity contour, and more broadly, it may be accompanied by a heavy sensation in the heart, and a bitter taste in the mouth. Therefore, emotion conceptualization is certainly multi-modal. Since CI children have poor reception of vocal characteristics, they may well construct their emotional concepts with little care for the auditory modality. As a consequence, the mental representations of emotions may not be as distinct as for normal-hearing children, and this could explain why CI children have deficits even in a facial emotion task. This is quite counter-intuitive considering that CI children might have more prominent visual emotion cues, one could almost hypothesize the contrary, namely that they should surpass NH kids at some point in their development, in a visual emotion task. But this does not seem to happen and this is what makes this field so interesting. The construction of emotional concepts is key.
There is an interesting parallel with respect to the concept of pitch, which I think is worth mentioning since your study is also about cross-language effects. In English, pitch is conceptualized (and taught) with descriptive such as “high” and “low”, which are easily confusable with spatial connotations, but also loudness connotations and emotional states. In other languages, French and Spanish, there are specific words for pitch, for example “aigu/grave”, and just this difference seems enough to induce a better ability to label changes in pitch early on (Costa-Giomi and Descombes, 1996). The way I understand this sort of phenomena is that concepts are easier to recognize when their mental representations are specific, more separated from each other in cognitive space. This is true of something as simple as pitch, as well as more complex notions like emotions.

In other words, what may be happening is that somehow, with advancements in cognitive skills, the CI children must catch up on their NH peers in emotional processing, sometime between 4-5 and 7-8 years of age, and I think this would deserve a close look. I'm glad the authors are already planning on such investigations, and I hope they will consider looking at pre-school children as well.

References:
Costa-Giomi, E., and Descombes, V. (1996). “Pitch labels with single and multiple meanings: A study with French-speaking children”, J. Res.Music Ed. 44, 204-214.
Wiefferink, C.H., Rieffe, C., Ketelaar, L., de Raeve, L., and Frijns, J.H.M. (2012). “Emotion understanding in deaf children with a cochlear implant”, J. Deaf Studies and Deaf Education, doi: 10.1093/deafed/ens042.

·

Basic reporting

In general, the literature is well-referenced in the paper. However, there are two places where it could be enhanced:

1. Line 38: Since this sentence is demonstrating the knowledge gap you wish to fill with this paper, it seems important that your reference is more recent than the 1986 Scherer review. Especially since you then go on to cite more recent studies.

2. Line 36: Would it be possible to get a more recent reference or two? 1994 is pretty old now. It would also strengthen your claim to have a few more example references demonstrating the breadth of studies that used visual or audiovisual stimuli.

The raw data is supplied and is very clear and seems totally anonymized.

The article conforms very well to PeerJ standards and is quite clear. The only possible issue is in the Reference section:

3. In the References section, it seems that the titles of the cited literature should all be lower case after the first letter. Check that you have done this. For instance, Line 202-203. Perhaps the rule is different for a book chapter title, this is unclear. Line 213-215 really should be lower case. Double check all references for this issue.

The figures in the paper are really wonderful, very clear and demonstrative. I only have one small suggestion for Table 1:

4. Table 1: Small note on formatting–consider centering the values and the column labels to make it a bit neater.

Overall, clear and professional English language is used throughout the paper. There are a few places where perhaps the sentence structure or word choice could be improved. I have outlined these instances below. Because there are a lot of small comments and none are particularly more important than the next, these are in line number order:

5. Lines 19-20: There may be a typo in this sentence (at least it showed up in other versions of the abstract). Written is “becrucial” instead of “be crucial”. Be sure the space needed between the words is actually there.

6. Line 21: The phrase, “…productions of multiple speakers…,” maybe should be, “…productions from multiple speakers…,” instead.

7. Line 29: I’m not sure if “furthermore” is really the right word here–maybe use the word “additionally” instead.

8. Lines 30-31: It is unclear if “emphasizes” is the right word in that sentence–maybe use the word “demonstrates” instead.

9. Line 32: The phrase, “age-related performances that are captured,” is a little awkward and unclear in meaning. It might be helpful to find another way to phrase the first half of that sentence.

10. Line 44: There does not need to be a comma between the words “than, when”.

11. Line 46: This phrase is a bit awkward, “… better in facial than vocal emotion recognition tasks”–maybe write, “better in facial emotion recognition tasks than in vocal emotion ones” instead.

12. Line 58: I’m a little confused by the statement, “may also lead to less robust representations of vocal emotion”–do you mean when they speak, they then use less robust representations of vocal emotion? Or that there are less robust representations of vocal emotion in their mind? It could be helpful to clarify the meaning of the phrase a bit.

13. Line 62-64: This sentence is a bit long and not clear in a few places. It could be broken into multiple shorter sentences for clarity (more specific suggestions are below in point 14 and 15).

14. Line 62: Regarding the beginning of the sentence, “This difference may…”–given the sentence before, don’t you mean “difficulty” instead of “difference”?

15. Line 63: The phrase, “… higher reliance on differences in speaking rate,” is awkward. Do you mean higher-than-average? Or a more weighted reliance? It would be good to clarify this by rephrasing.

16. Line 67: It would be good to add “audiovisual” to the sentence since you mention videos, i.e. “has used visual or audiovisual materials…”.

17. Line 68-70: I recommend breaking that sentence into two sentences, it’s a run-on as-is.

18. Line 78-79: Use either text or numbers to describe how many participants there were because mixing makes the sentence confusing at first glance.

19. Line 82: The phrase, “approved by local ethics committees…,” is missing a “the”: should be “approved by the local ethics committees”.

20. Line 105-107: This sentence is a bit awkward. Perhaps rephrase, “Since the present experiment… data were collected from the same population of children and adults for multiple experiments…”

21. Line 158: For the phrase, “as was shown for the development…”, recommend replacing “for” with “in research on”: so phrase as, “as was shown in research on the development…”

22. Line 160-161: That comma is not needed–maybe use the word “since” instead of “as”.

23. Line 170: Clarify that this sentence refers specifically to your results: i.e. “… the lack of a native language benefit in our results may also be due…”

24. Line 176: You might want to rephrase the sentence as follows, “… for typically-developing, school-age children with normal hearing…” just to break up all those descriptors and make it easier to read.

25. Line 176-179: Break that sentence into two sentences, because they are not totally related ideas and it is a bit awkwardly phrased.

26. Line 180-184: These two sentences feel redundant of each other and also don’t really provide a clear point. Is your point that you are now expanding this work into the CI population? It would be good to make that clearer.

27. Line 188: You are missing the word “the”: It should be, “We are grateful to all of the children, parents…”

28. Line 188-189: I recommend replacing the word “of” with “recorded for”: so, “…the speakers recorded for our stimuli, and…”

Experimental design

The work is original and fits well within the scope of this journal. The research questions are generally well defined and are highly relevant. There is just one place in the paper where two sentences setting up the research question might be rephrased for further clarity:

29. Lines 17-20: The first and second sentence of the abstract do not quite connect. I understand you are setting up the issues you wanted to address with your paper, but connecting the thoughts better would make that clearer. Maybe add on to the end of the first sentence “or of good audio quality” to connect them.

The investigation in this study is well and ethically performed. I only had one question about the methodology:

30. Line 168: Why did you decide to use only three basic emotions for this test? The decision is not explained in the paper. It could strengthen the work to address this choice in the introduction with a sentence or two.

The methods are largely described in enough detail to be replicated. There is just one place where things are a little bit unclear:

31. Lines 100-102: These sentences are a little difficult to parse, particularly the phrase “combinations of two times one sentence and one time the other sentence”. Maybe write, “stimuli with three productions (one sentence repeated once + another sentence) per emotion and per speaker (3 productions x 3 emotions x 4 speakers) as well as 4 practice stimuli with one production per speaker…”. The use of the word “items” is confusing because it wasn’t used before–maybe continue using the word “productions” instead. Also for the sake of replication, it would be helpful to know which sentence was repeated and which was not.

Validity of the findings

All the data for the project are provided and very clear. The results are clearly laid out with an excellent figure. The conclusions are fairly well-stated. There are only two instances where things could be clearer:

32. Lines 26-29: This sentence is a bit confusing mainly because it is too long and changes focus too often. I recommend that you split it into two sentences: one about how the children improved, and one sentence about how they still did not reach adult-like values.

33. Line 184-186: It’s not immediately clear why the EmoHI Test is suitable for hearing-impaired or clinical populations specifically, as your sentence currently implies. Perhaps break it into two sentences. The first about how the results indicate the usefulness of EmoHI across ages and languages. Then a second sentence about how this normative data and the high quality of the recordings make EmoHI also useful for hearing impaired and other clinical populations.

Additional comments

Overall this is a really well-done, well-written article that, with minor changes, seems ready to be published. Congratulations to the authors on producing such a concise, informative, and valuable paper.

---

## Round 0.2 · Minor Revisions

Thanks for your careful revision. I invite you to provide the last minor revisions required by Reviewer 2.

·

Basic reporting

Yes, it is very well written. The authors have added several references, including more recent papers. The literature is well covered. The table is a nice addition. Raw data are shared. All good.

Experimental design

As I mentioned in my earlier comments, the study was well conceived and executed. The research question is well defined and the results clear and concisely reported. Rigorous approach and ethical standards.

Validity of the findings

The findings are interesting and the EmoHI would be a very useful tool for future investigations. I think the developmental trend is particularly worth examining further.

Additional comments

The authors have done a good job at revising their manuscript.
I have no more comment. Congratulations on a great contribution to the field!

·

Basic reporting

Overall, the manuscript is quite clear and cites an appropriate amount of related work throughout it. However, especially in some of the newer sections, there are still some unclear/awkward sentence structures. I have included some corrections and suggestions below:

- Line 22: I would add “are not” to make the sentence even clearer, i.e. “… test materials are not always readily available or are not of good quality…”
- Line 37-38: It is not clear how exactly implantation age was “taken into account”. Maybe instead: “…when measuring from their age at CI implantation (hearing age)”.
- Line 38-40: This sentence doesn’t quite work. Something clearer might be: “However, CI children showed great variability in their performance, ranging from ceiling (97.2%) to below chance-level performance (27.8%), which could not be explained by chronological age alone.”
- Line 50-53: The sentence is not properly constructed in English, the second half of it at least. Something like this would be more correct: “… it seems to take until late childhood for this ability to develop to adult-like levels”
- Line 83-84: This sentence is unclear. Effects of what conceptual knowledge, emotional concepts? How is that related to pitch discrimination?
- Line 85-87: Put the word “also” later in the sentence, i.e.: “Finally, perceptual limitations, such as increased F0 discrimination thresholds (source), may also play a role…”
- Line 102: Is “linguistic distance” the same as geographical distance? Maybe include a very brief definition in parentheses.
- Line 112-113: The last part of this sentence feels random here, like it should have been mentioned earlier. Because in this paragraph, the main point is why good quality audio stimuli are important for testing hearing-impaired participants, not that their ability to hear emotion from voices is important for their quality of life. This part specifically seems unrelated at this point in the article: “… which have been shown to relate to their self-reported quality of life”
- Line 238-9: The way the sentence is worded is a bit awkward and unclear. Perhaps something like: “Considering CI children’s hearing age instead of chronological age, 9 out of 14 CI children (64.3%) still show performance within…”
- Line 287: The sentence is a bit clunky, it would help to add “and indicates that” in the middle, i.e.: “…level, which shows that almost all CI children could reliably perform the task, and indicates that the task seems…”
- Line 290: Instead of “even”, use “still” later in the sentence. i.e.: “… instead of their chronological age, 9 out of 14 CI children still fell within…”
- Line 304-307: This sentence is a bit confusing at the end. What is meant by “an acute effect” and “a long-term effect”? Basically the listing here is confusing. By the end of the sentence, it isn’t clear what you are saying CI children’s performances are related to.
- Line 319: The use of the word “evidence” as a verb is a bit off-putting/confusing at first glance. Perhaps use “demonstrate” or something else.
- Figure 2 & 3: Perhaps explain the dotted line thresholds in the caption, it isn’t totally clear what that represents.

Experimental design

No corrections here, everything is much clearer post-revisions. Well done!

Validity of the findings

No comments here as well, looks good!

Additional comments

The revisions were nicely done as a whole. There really are only some writing adjustments that could be made for clarity. Otherwise, the article seems to be in really good shape. It was interesting to read the CI children results, I support your decision to include them. Well done as a whole!

---

## Round 0.3 · accepted · Accept

I am happy to inform you that you paper has been accepted for publication on PeerJ.